# The Influence of Consortia of Beneficial Microorganisms on the Growth and Yield of Aquaponically Grown Romaine Lettuce

Lidia Sas-Paszt *[iD], Paweł Trzciński [iD], Anna Lisek [iD], Sławomir Głuszek [iD], Bożena Matysiak [iD] and Stanisław Kaniszewski [iD]

The National Institute of Horticulture Research, Konstytucji 3 Maja 1/3 Str., 96-100 Skierniewice, Poland
* Correspondence: lidia.sas@inhort.pl

**Abstract:** This study evaluated the effects of fish farm wastewater from the production of hybrid sturgeon (*Acipenser gueldenstaedtii* Brandt × *Acipenser baeri* Brandt) on the growth and quality parameters of romaine lettuce (*Lactuca sativa* var. longifolium cv. "Elizium"). The tested combinations were fish farm wastewater, fish farm wastewater enriched with one of the three microbiological consortia, and fish farm wastewater supplemented with minerals. The best growth parameters of romaine lettuce plants were obtained in the combination of wastewater from fish farming supplemented with mineral nutrients. The application of fish farm wastewater and beneficial microbiological consortia positively influenced the fresh weight of lettuce leaves and the number of leaves per plant. However, plants fed with wastewater supplemented with minerals were characterized by the strongest symptoms of leaf tip-burn and the lowest commercial value. By comparison, plants fed only with fish farm wastewater or wastewater with microorganisms were characterized by a high, similar commercial value. After the application of increased doses of minerals, there was evidence of greater activity of microorganisms involved in nutrient cycling in aquaponic lettuce cultivation. The application of the microbiological consortia and minerals significantly increased the numbers and activity of the bacteria in the culture liquids 7, 14, and 21 days after inoculation.

**Keywords:** hybrid sturgeon; romaine lettuce; aquaponics system; microbiological consortia; fish farm wastewater; activity of the bacteria





## 1. Introduction

Traditional aquaculture is growing rapidly and is the most dynamic segment of industrial animal production [1,2]. However, the development of this activity is associated with an extremely high demand for water, which is increasingly difficult to meet in conditions of increasing water shortage [3,4]. At the same time, aquaculture has a large impact on the environment, contributing to the growing pollution of water reservoirs into which post-farming sewage is discharged. The wastewater discharged from growth tanks contains dissolved minerals, such as nitrogen (N) and phosphorus (P), numerous specific organic and inorganic compounds, and suspended matter, which is mainly derived from unused feed and fish excreta [5–9]. This problem applies particularly to flow-through systems, where water flows through breeding tanks [10]. These wastes must be removed from the farming systems as they have a negative impact on fish growth and survival [11,12]. A compound harmful to farmed fish is ammonia ($NH_3$) dissolved in water, to which many fish species are sensitive [13–15]. This compound lowers the fish's appetite and food intake, resulting in reduced growth [16]. Therefore, the development of aquaculture requires the development and application of new technologies that will make use of wastewater and nutrients in other processes.

At the same time, a major problem in plant production is the lack of clean water and the gradual loss of agricultural land, which leads to a progressive decline in agricultural production in many regions of the world [17]. The implementation of hydroponic and

aquaponic techniques, particularly various vertical farming systems, allows a significant increase in plant productivity per square meter [18,19]. Another threat to crop productivity is the limited availability of mineral fertilizers and their prices. The prices, in particular, are a limiting factor as they have increased significantly in recent years. Therefore, new sources of mineral nutrients are being sought, and cultivation technologies are being developed to improve the efficiency of the use of nutrients in agricultural crops.

One of the ways to reduce the effects of the above threats is the implementation of a technology that integrates aquaponic animal husbandry with the cultivation of plants, in particular, vegetables [20].

The use of wastewater from aquaculture in the cultivation of plants not only prevents its direct discharge into the environment but also allows it to be purified and recirculated in aquaculture [21,22]. Therefore, aquaponics with recirculating water circulation is the answer to the problems related to the dynamic development of the demand for fish and other aquatic organisms (e.g., seafood) in the world [23,24]. Recirculating aquaponic systems integrate hydroponic plant production and aquaculture production of fish and other organisms. Systems of this type minimize the need to supply and replace fresh water in the production system, resulting in significant water savings [20,23,25–28]. The quality and nutritional value of the fish produced in recirculating aquaponics are similar to the quality of non-farmed fish; however, the production itself must be carried out very carefully to avoid potential problems with the quality of fish meat resulting from, among other factors, the inappropriate breeding parameters [29].

Benefits are also obtained in the case of the plants produced. In indoor aquaponic cultivation, the pressure of pests and diseases is reduced, which lowers the number of necessary plant protection treatments and reduces the use of plant protection products [30]. Fast-growing vegetables with edible leaves, such as spinach [13,23,31,32], lettuce [13,24,33–36], celery, and herbal plants [37], are perfect for growing in aquaponics. In addition, vegetables with edible fruits, such as tomato [26,38,39], cucumber [40–42], pepper [43,44], and pumpkin [45], can be grown in this system. Depending on the plant species used in the system, different effectiveness of water purification from mineral components is obtained. Effective in purifying water are cruciferous (brassica) vegetables, which have a high demand for mineral nitrogen [46].

Romaine lettuce is one of the species perfectly suited to cultivation in fully controlled conditions, especially in soilless cultivation. Such technologies are hydroponics and aquaponics, with the latter gaining increasing popularity. In aquaponic cultivation, water circulates in a closed circuit. Wastewater from fish farming is used to irrigate and fertilize cultivated plants. The plants take up the mineral compounds, and the wastewater purified by the plants returns back to the breeding tanks. As a result, the demand for water in the recirculating aquaponic system is minimized, and the disposal of waste from fish production to the environment is limited.

Among the diversity of plant diseases occurring in aquaponics, soil-borne pathogens, such as *Fusarium* spp., *Phytophthora* spp., and *Pythium* spp., are the most problematic due to their preference for humid/aquatic environment conditions. Phytophthora spp. and *Pythium* spp., which belong to the Oomycetes pseudo-fungi, require special attention because of their mobile form of a dispersion, the so-called zoospores that can move freely and actively in liquid water [47]. High nutrient concentrations, especially in moist, warm environments, such as greenhouses, facilitate the growth of phytopathogens, such as *Phytophthora oomycetes*, which can spread rapidly in circulating water and can cause plant die-offs [48,49]. For this reason, research on the ability of the aquaponic biotope to inhibit diseases should be intensified, as well as the isolation, characterization, and formulation of antagonists of microbial plant pathogens [47].

Leafy vegetables, such as romaine lettuce, are perfect for growing in hydroponic and aquaponic systems because their production cycle can be as short as 3–4 weeks, and their nutritional requirements are relatively small. There are also relatively fewer problems with pests and diseases in such a short production time compared to vegetables requiring a

long production cycle [50–52]. In addition, plants grown in aquaponic systems do not accumulate large amounts of minerals such as nitrates [53].

Compared to cultivation in the soil, hydroponics, and in particular aquaponics with recirculating water circulation, can be more profitable and can provide additional benefits, such as a significant reduction in the use of production inputs and reduced environmental pollution [54]. Aquaponics can be used as a method or system to reduce the need for fresh water, and fish, vegetables, or herbs can be produced together to close the water loop in the production system [3,24,34,39]. In addition, farming in a strictly isolated environment makes it possible to reuse the carbon dioxide emitted by the fish to feed the plants. Thanks to this supplementation with $CO_2$, not only is a higher yield obtained but the carbon footprint of fish and vegetable production in aquaponics is also reduced.

The effectiveness of water purification and improvement in mineral utilization in aquaponic systems can be increased by supplementing the cultivation system with beneficial microorganisms. In the aquaponic system, ammonia is converted by nitrifying bacteria to nitrate ions and, in this form, is taken up by plants [55,56]. The use of bacteria allows a significant reduction in the concentration of nitrogen compounds, both ammonium and nitrate forms, in the water in which fish are bred [36,57]. In addition, the bacteria break down organic compounds and fish excreta in the water, thus increasing the availability of other nutrients [58,59]. The effectiveness of biological purification is increased by using special bioreactors or beds in which the flowing water is purified by microorganisms [60,61].

There have been very few studies investigating the effect of using beneficial microorganisms to enrich fish farm wastewater fed to plants on the growth of those plants. Usually, studies on the use of microorganisms have been carried out in hydroponic systems. The application of microorganisms to cultivation vessels or directly to the nutrient solution improves the growth and yield of hydroponically cultivated plants [62,63] and increases their resistance to pathogens [64,65]. On the other hand, the use of bacteria able to fix atmospheric nitrogen allows the use of reduced doses of fertilizers in hydroponics without any loss in yield [66]. Beneficial microorganisms used in a hydroponic system can also improve the quality of the crop obtained, e.g., by reducing the nitrate content in the crop [67]. In hydroponic crops, microorganisms of the genus Bacillus and/or Pseudomonas have an inhibitory effect on plant pathogenic microorganisms by competing for nutrients, inducing plant defense mechanisms, and by their antagonistic activity through the production of antifungal and/or antibacterial substances, such as cell wall degrading enzymes, bacteriocins and antibiotics, and lipopeptides [47].

The romaine lettuce (*Lactuca sativa* L. var. longifolia) is a vegetable liked by consumers due to its organoleptic qualities, in particular its visual qualities, interesting taste and aroma, and crunchy texture of the leaves [68]. The fish used in the study is a cross between the Russian sturgeon (*Acipenser gueldenstaedtii* Brandt) and the Siberian sturgeon (*Acipenser baerii* Brandt). It is a breeding (not natural) hybrid of the F1 generation of these two species with a similar number of chromosomes [69]. Aquaponics with sturgeons and their hybrids have been the subject of a number of studies, with the main focus on sturgeon productivity and meat quality and less focus on the efficiency of crop production [33,70].

The purpose of this study was to investigate the growth performance of romaine lettuce grown in a recirculating aquaponic system with hybrid sturgeon. Another aim of the research was the selection of bacterial strains intended to stimulate the yielding of romaine lettuce grown in an aquaponic technology of sturgeon fish farming.

## 2. Materials and Methods

### 2.1. Selection of Beneficial Bacterial Strains for Water Enrichment in Aquaponic Cultivation of Romaine Lettuce

The biological material used to obtain bacterial isolates was fish excreta, skimmer sludge, and romaine lettuce rhizosphere soil. The bacterial isolates were obtained by serial dilution on agar media. Characterization of the obtained bacterial isolates was carried out by determining such properties as oxidation of $NH_4$ ions by assessing the

growth of bacterial colonies on a mineral medium containing $NH_4Cl$ as a source of nitrogen (https://www.dsmz.de/microorganisms/medium/pdf/DSMZ_Medium1583.pdf (accessed on 12 January 2021), and antibiosis properties in relation to inhibiting the pathogen Phytophthora cactorum. The production of secondary metabolites toxic to phytopathogenic fungi was assessed by the double-culture method. Plates with agar media (potato dextrose agar and soil agar) were inoculated with a culture of *P. cactorum* in the center and with the test bacteria at the edge. After inoculation, the plates were incubated for 14 days at 26 °C. Bacterial isolates that inhibited the growth of *P. cactorum* were judged as producing toxic secondary metabolites against the pathogen. Identification of the isolated bacterial strains was based on the analysis of the 16SrRNA gene sequence and comparison of the obtained sequences with NCBI (National Center for Biotechnology Information) data https://blast.ncbi.nlm.nih.gov/Blast.cgi?PROGRAM=blastn&PAGE_TYPE=BlastSearch&LINK_LOC=blasthome (accessed on 18 February 2021). In addition, biochemical identification of the selected bacterial strains was carried out using the BIOLOG system.

### 2.2. Crop and Experimental Site

Romaine lettuce *Lactuca sativa* var. longifolium cv. "Elizium" (Enza Zaden)—a mini type was cultivated in a free-standing outdoor container (Weldon, Brzezówka, Poland) with the dimensions of $6.0 \times 2.6 \times 3.2$ m adapted to a phytotron by BIOSELL (Warsaw, Poland) fitted with two two-shelf racks, as described previously [71]. Fourteen-day-old seedlings produced in rockwool cubes (ROCKWOOL Polska Sp. z o.o., Poland) were used in the study. Plants were grown in polystyrene boxes with a capacity of 20 L filled with wastewater from fish (sturgeon) farming or nutrient solution. There were six plants in each box, which were mounted on floating polystyrene rafts (24 plants per $m^2$). The temperature in the phytotron was set at 20/18°C day/night, and the relative air humidity was set at 65%. For illumination, LED lamps with a light spectrum of 70:18:12 RGB with an intensity of 160 $\mu mol\ m^{-2}\ s^{-1}$ and photoperiods of 20 h were used.

The following treatments were used in the experiment: (1) control (fish farm wastewater); (2) fish farm wastewater + beneficial microorganism consortium I (BM I); (3) fish farm wastewater + beneficial microorganism consortium II (BM II); (4) fish farm wastewater + beneficial microorganism consortium III (BM III); and (5) fish farm wastewater supplemented with mineral nutrients (MN).

Before setting up the experiment, the wastewater from the fish farm was analyzed. The concentrations of nutrients are presented in Table 1.

**Table 1.** Concentration of nutrients in the wastewater from fish farming (mg $L^{-1}$).

| Nutrient | Concentration |
|---|---|
| $N\text{-}NO_3^-$ | 62.0 |
| $N\text{-}NH_4$ | 7.3 |
| $P\text{-}PO_4^{-3}$ | 5.68 |
| $K^+$ | 17.6 |
| $Ca^{+2}$ | 138 |
| $Mg^{+2}$ | 27.6 |
| $Na^+$ | 18.9 |
| $Cl^-$ | 50.1 |
| $SO_4^{-2}$ | 130 |
| Fe | 0.06 |
| Fe total | 0.29 |
| Mn | 0.02 |
| Cu | 0.02 |
| Zn | 0.10 |
| B | 0.11 |

In the treatment with fish farm wastewater supplemented with mineral nutrients, the amounts of macroelements and microelemnts in the solution were supplemented to the

following levels: N-NO$_3$—130; N-NH$_4$—11; P—40; K—180; Ca—200; Mg—35; Fe—2.0; Mn—0.76; Zn—0.16; B—0.32; Cu—0.16; and Mo—0.04 (mg dm$^{-3}$). EC was 1.6 mS cm$^{-1}$, and pH was 6.0 [71].

To maintain the concentrations of mineral nutrients in the containers, the fish farm wastewater and the mineral nutrient solution were changed weekly. Additionally, the prepared consortia of microorganisms were added to the fish farm wastewater at each exchange. Microbial inocula were prepared by culturing the bacteria for 72 h in the Murashige and Skoog medium [72] (2.93 g of medium per 1000 g of water) with the addition of glucose (0.01%) and sucrose (0.01%). Next, the suspensions prepared in this way were applied to the liquid in which the lettuce plants were grown, in the amount of 500–750 mL per container (with a capacity of about 21 L). The pH was determined with the potentiometric method and EC using the conductivity method. Mineral components in the nutrient solution, such as N-NO$_3$, were analyzed by the potentiometric method; P, K, Ca, Mg, and SO$_4$ were analyzed by the spectrophotometric method using a sequential emission spectrometer with inductively coupled plasma (ICP Perkin-Elmer model Optima 2000 DV, Boston, MA, USA).

Plants were harvested after 30 days of cultivation by cutting off the leaves above the collar. Morphological traits include fresh and dry weight of leaves, plant height, its diameter, head circumference, the number of leaves per plant, the number of leaves with tip-burn, the commercial value on a scale of 1 to 5 (1—the worst, and 5—the best quality); the features of the root system were also determined. Using the EPSON EXPRESSION 10000 XL root scanner and WinRhizo version 2009c software (Regent Instruments Inc., Quebec City, QC, Canada), root characteristics, such as fresh and dry weight, root length, root surface area, root diameter, root volume, and the number of root tips, were determined.

*2.3. Microbiological Analysis of the Media for Growing Lettuce Plants*

Microbiological analyses (determination of the numbers of bacteria and estimation of their activity and biodiversity) were performed 48–72 h after the application of bacteria and at 7-day intervals after inoculation. The population of microorganisms was determined by bottom plating serial ten-fold dilutions using the R2A agar medium. The activity and biodiversity of the microorganisms present in the plant growth medium were determined using the BIOLOG system and EcoPlate plates. The activity of microorganisms was estimated on the basis of Average Well Color Development (AWCD) [73]. The microbial activity coefficient was calculated according to the formula:

$$AWCD = \Sigma\ OD_i/31$$

where OD$_i$ is the optical density of individual wells.

Microbiological diversity was estimated using the Shannon–Weaver coefficient (H) H $= -\Sigma\ p_i(\ln p_i)$, where $p_i$ is the level of microbial activity in each well (OD$_i$) divided by the activity in all the wells ($\Sigma$ OD$_i$) [74]. When assessing the level of activity of microorganisms for the "H" coefficient and the amount of metabolized substrates, the threshold value OD = OD$_i$ − OD$_{(control\ well)}$ was established.

*2.4. Statistical Analysis*

Statistical analysis was performed using a one-way analysis of variance using Tukey's test, $\alpha$ = 0.05, in the Statistica 13.1 statistical program.

**3. Results**

*3.1. Selection of Beneficial Bacterial Strains and Development of Microbial Consortia*

As a result of the tests on 153 bacterial isolates, six strains were selected with the most favorable traits related to the antagonism towards *Phytophthora cactorum* and the ability to oxidize ammonium ions, which are toxic to fish. The selected bacterial isolates were obtained from fish excreta (RXAAC *Klebsiella* spp., RXBAB *Klebsiella* spp.), skimmer sludge

(ODBA *Nocardia* spp., ODBB *Rhodococcus* spp.), and rhizosphere soil of romaine lettuce (SpBX2020 *Paenibacillus polymyxa*, SpBY2020 *Paenibacillus polymyxa*). Based on the analysis of the 16S rRNA gene sequence, the selected bacterial isolates were found to belong to the genus *Klebsiella* (RXAAC, RXBAB) and to the species *Paenibacillus polymyxa* (SpBX2020, SpBY2020) (Table 2). On the basis of the biochemical profile obtained using the BIOLOG system, the strain "ODBA" was assigned to the genus Nocardia spp. and the strain ODBB to the genus *Rhodococcus* spp. The selected strains showed the properties of producing biofilm and biomass in the presence of ammonium chloride: RXAAC (*Klebsiella* spp.), RXBAB (*Klebsiella* spp.), ODBA (*Nocardia* spp.), ODBB (*Rhodococcus* spp.), or the ability to inhibit the pathogen *Phytophthora cactorum*: SpBX2020 (*Paenibacillus polymyxa*), SpBY2020 (*Paenibacillus polymyxa*).

**Table 2.** Identification of microorganism strains based on the similarity of the 16S rRNA gene sequence to the sequences stored in the NCBI database.

| Strain | The Genus/Species of Bacteria with the Greatest Similarity to the NCBI Sequence | NCBI Sequence No. | Degree of Similarity (%) | Identification |
|---|---|---|---|---|
| RXAAC | *Klebsiella* sp. strain KKP_3088 | MT580115.1 | 99.04 | *Klebsiella* sp. |
| RXBAB | *Klebsiella* sp. strain D8 | MT580115.1 | 98.98 | *Klebsiella* sp. |
| SpBX2020 | *Paenibacillus polymyxa* strain DSM 36 | NR_117733.2 | 99.8 | *Paenibacillus polymyxa* |
| SpBY2020 | *Paenibacillus polymyxa* strain NBRC 15309 | NR_112641.1 | 99.63 | *Paenibacillus polymyxa* |

Based on the assessment of the properties of the bacterial strains, three microbiological consortia were developed for the aquaponic cultivation of romaine lettuce:

- Beneficial microorganism consortium I containing strains with the properties of $NH_4^+$ ion oxidation and growth on a mineral medium containing $NH_4Cl$ as a nitrogen source: RXAAC (*Klebsiella* spp.); and RXBAB (*Klebsiella* spp.);
- Beneficial microorganism consortium II containing strains with the properties of inhibiting the pathogen *Phytophthora cactorum*: SpBX2020 (*Paenibacillus polymyxa*); and SpBY2020 (*Paenibacillus polymyxa*);
- Beneficial microorganism consortium III containing strains with the properties of $NH_4^+$ ion oxidation and growth on a mineral medium containing $NH_4Cl$ as a source of nitrogen: ODBA (*Nocardia* spp.), ODBB (*Rhodococcus* spp.).

*3.2. Evaluation of the Growth of the Above-Ground Part and Roots of Romaine Lettuce*

The best growth parameters of romaine lettuce plants were obtained in combination with wastewater from fish farming being supplemented with mineral nutrients (Figure 1). Next in the sequence were, respectively: fish farm wastewater + beneficial microorganism consortium I; fish farm wastewater + beneficial microorganism consortium II; fish farm wastewater + beneficial microorganism consortium III; and the control (fish farm wastewater). However, the plants fed with water supplemented with minerals were characterized by the strongest symptoms of leaf tipburn and the lowest commercial value. In contrast, the plants fed with fish farm wastewater only or wastewater with microorganisms were characterized by a high, similar commercial value. The application of minerals increased the diameter of romaine lettuce roots (Table 3). Moreover, after the application of the microbial consortia or minerals, a reduction in the number of plant root tips was observed.

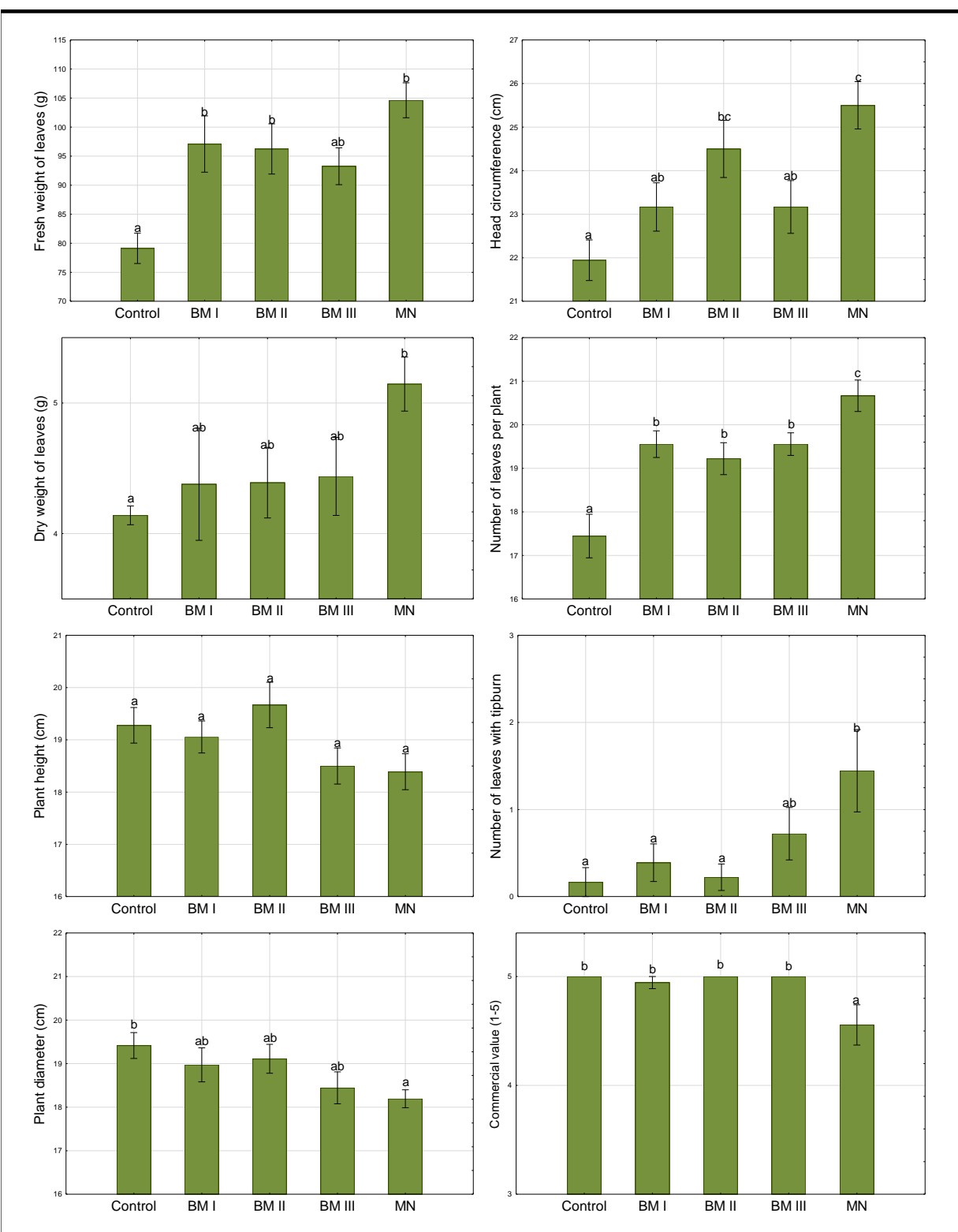

**Figure 1.** Fresh and dry weight of leaves, plant height, plant diameter, head circumference, number of leaves per plant, number of leaves with tipburn, and commercial value of "Elizium" romaine lettuce grown in fish farm wastewater (control), fish farm wastewater supplemented with beneficial microorganism consortia I, II, and III (BMI, BM II, BM III), or fish farm wastewater with mineral nutrients (MN). Bars represent means ± SE. Means followed by the same letter are not significantly different ($p < 0.05$) according to Tukey's HSD test.

**Table 3.** Effect of fish farm wastewater, mineral medium based on fish farm wastewater, and microbiological consortia on romaine lettuce root growth characteristics.

| Combination | Root Fresh Weight [g] | Root Dry Weight [g] | Root Length [cm] | Root Surface Area [cm²] | Root Diameter [mm] | Root Volume [cm³] | Number of Root Tips |
|---|---|---|---|---|---|---|---|
| Control (fish farm wastewater) | 7.44 ± 0.5 a | 0.27 ± 0.02 a | 4315.7 ± 913.4 a | 545.5 ± 74.9 a | 0.43 ± 0.04 a | 6.1 ± 0.29 a | 5016 ± 941.5 b |
| Fish farm wastewater + beneficial micro-organisms Consortium I | 7.19 ± 1.7 a | 0.25 ± 0.06 a | 3573.8 ± 744.8 a | 535.6 ± 109.9 a | 0.47 ± 0.01 a | 6.4 ± 1.29 a | 3283 ± 623 a |
| Fish farm wastewater + beneficial micro-organism Consortium II | 7.30 ± 0.3 a | 0.24 ± 0.02 a | 4365.3 ± 1755.0 a | 555.8 ± 130.1 a | 0.43 ± 0.08 a | 5.8 ± 0.3 a | 3480 ± 107.5 a |
| Fish farm wastewater + beneficial micro-organism Consortium III | 9.34 ± 2.9 a | 0.31 ± 0.09 a | 3749.4 ± 233.4 a | 443.4 ± 38.6 a | 0.50 ± 0.05 ab | 7.6 ± 2.01 a | 3497 ± 436.5 a |
| Mineral medium based on fish farm wastewater | 9.59 ± 1.1 a | 0.35 ± 0.05 a | 2126.1 ± 206.6 a | 595.6 ± 98.3 a | 0.66 ± 0.01 b | 7.4 ± 0.57 a | 2311 ± 114.5 a |

Statistical analysis was performed using one-way analysis of variance using Tukey's test, $\alpha = 0.05$, using the Statistica 13.1 program. Means with the same letter are not significantly different from each other.

### 3.3. Microbiological Analysis of the Media for Growing Lettuce Plants

Following the application of the microbial consortia and minerals, significantly higher bacterial activity was observed after 48–72 h and after 7, 14, and 21 days from the application of the biopreparations (Table 4). Immediately after the application of the consortia, the highest activity was found in the liquid treated with Consortium I, whereas 7–28 days after the application, the highest microbial activity was found in the samples of the liquid treated with minerals. Increased bacterial biodiversity in the culture liquids was observed only 48–72 h after the application of the microbial consortia or mineral nutrients. The application of the microbiological consortia or minerals contributed to an increase in the number of bacteria in the culture liquids, with the highest numbers of bacteria observed after the application of Consortium I and Consortium II 48–72 h after inoculation.

**Table 4.** Effect of the application of microbiological consortia and mineral nutrients on the abundance ($\times 10^5$ CFU $\times$ mL$^{-1}$), activity (AWCD), and biodiversity (Index H) of the microorganisms in the plant growing media.

| Combination | 48–72 h after Consortium Application | | | 7 Days After Consortium Application | | | 14 Days after Consortium Application | | | 21 Days after Consortium Application | | | 28 Days after Consortium Application | | |
|---|---|---|---|---|---|---|---|---|---|---|---|---|---|---|---|
| | AWCD | Index H | $\times 10^5$ CFU $\times$ mL$^{-1}$ | AWCD | Index H | $\times 10^5$ CFU $\times$ mL$^{-1}$ | AWCD | Index H | $\times 10^5$ CFU $\times$ mL$^{-1}$ | AWCD | Index H | $\times 10^5$ CFU $\times$ mL$^{-1}$ | AWCD | Index H | $\times 10^5$ CFU $\times$ mL$^{-1}$ |
| Control (fish farm wastewater) | 0.17 ± 0.06 a | 2.11 ± 0.07 a | 1.24 ± 0.14 a | 0.33 ± 0.01 a | 2.69 ± 0.07 a | 2.54 ± 0.1 a | 0.23 ± 0.01 ab | 2.24 ± 0.3 a | 1.64 ± 0.32 a | 0.19 ± 0.01 a | 2.54 ± 0.46 a | 1.58 ± 0.17 a | 0.29 ± 0.04 a | 2.66 ± 0.13 a | 2.07 ± 0.19 a |
| Fish farm wastewater + beneficial micro-organisms Consortium I | 1.01 ± 0.13 d | 3.11 ± 0.16 b | 10.5 ± 2.05 c | 0.62 ± 0.1 c | 2.96 ± 0.03 a | 5.17 ± 0.52 c | 0.13 ± 0.01 a | 2.07 ± 0.21 a | 1.18 ± 0.05 c | 0.93 ± 0.01 b | 2.85 ± 0.07 a | 8.45 ± 0.34 b | 0.57 ± 0.02 bc | 2.79 ± 0.25 a | 4.38 ± 0.13 b |
| Fish farm wastewater + beneficial micro-organism Consortium II | 0.92 ± 0.12 cd | 2.77 ± 0.08 b | 9.03 ± 1.95 c | 0.45 ± 0.1 b | 3.15 ± 0.54 a | 4.09 ± 0.33 b | 0.31 ± 0.02 b | 2.72 ± 0.22 a | 2.43 ± 0.54 b | 0.28 ± 0.04 a | 2.47 ± 0.15 a | 2.0 ± 0.14 a | 0.44 ± 0.09 b | 2.9 ± 0.29 a | 4.89 ± 0.68 b |
| Fish farm wastewater + beneficial micro-organism Consortium III | 0.44 ± 0.13 ab | 2.95 ± 0.02 b | 3.38 ± 0.47 ab | 0.59 ± 0.01 c | 3.25 ± 0.22 a | 4.54 ± 0.27 ab | 0.12 ± 0.02 a | 2.71 ± 0.23 a | 1.2 ± 0.07 ab | 0.8 ± 0.26 b | 2.98 ± 0.32 a | 8.89 ± 1.16 b | 0.63 ± 0.01 c | 2.67 ± 0.21 a | 4.85 ± 0.15 b |
| Mineral medium based on fish farm wastewater | 0.67 ± 0.05 bc | 2.76 ± 0.3 b | 5.11 ± 0.77 b | 0.99 ± 0.02 d | 3.19 ± 0.12 a | 6.6 ± 0.53 bc | 0.56 ± 0.05 c | 2.72 ± 0.24 a | 3.73 ± 0.34 bc | 0.15 ± 0.02 a | 2.6 ± 0.4 a | 1.36 ± 0.05 a | 0.68 ± 0.04 c | 2.9 ± 0.38 a | 7.56 ± 0.53 c |

The results of microbiological analyses were verified with univariate analysis of variance using Statistica 10. Homogenous groups were determined with the HSD test for $p = 0.05$. Means with the same letter are not significantly different from each other.

## 4. Discussion

Aquaponics in a closed circulation system is an excellent way to effectively use water and minerals when farming fish and other aquatic organisms in the cultivation of plants, especially fast-growing leafy vegetables, and herbs. One of the main reasons behind the development of aquaponic techniques is the reduction in environmental pollution. In traditional aquaculture, wastewater with the excreta of fish or other cultured organisms ends up in the environment as waste or requires costly treatment. On the other hand, it is necessary to constantly replenish growing tanks with fresh water, which means that aquaculture competes for its resources with the living needs of humans, with other types of animal husbandry, agriculture, and industry.

In aquaponics, the water purified by the plants is returned to the fish production systems and not to the environment. Consequently, it is possible to reduce or even eliminate the costs of purifying wastewater. On the other hand, a closed system and the use of excreta from farming fish and other aquatic organisms to fertilize plants allows us to substitute the mineral fertilizers necessary in hydroponic production or at least reduce their amount.

The efficiency of aquaponic growing of crops is comparable to that obtained in traditional hydroponic cultivation based on optimal doses of mineral fertilizers in the medium, although it usually depends on the species of the cultivated plant. For example, Pantanella et al. [75] obtained a lettuce yield from aquaponic cultivation with the Nile tilapia (*Oreochromis niloticus* L.) at the level of 2.7 kg m$^{-2}$ compared to 2.8 kg m$^{-2}$ from hydroponic cultivation. The plants from the aquaponic cultivation were characterized by lower phosphorus content, but at the same time, they contained more calcium, potassium, magnesium, and sodium. Purwandari et al. [76] observed that romaine lettuce grown in a hydroponic system was characterized by better growth compared to plants grown in an aquaponic system fed only with wastewater from farming the giant gourami (*Osphronemus goramy*). However, in that case, the main purpose of the experiment was to check how the fish, grown in the water purified by plants, behaved. At the same time, the authors noted that fish reared in aquaponic systems with water circulation were characterized by a greater weight than fish reared in a traditional hydroponic system. A similar result regarding the effect of water purification by plants on the weight of pearl gourami (*Trichogaster leerii*) was also obtained by Makhdom et al. [38]. In that case, they used cherry tomato plants to purify the wastewater. The highest weight of fish, their survivability, and also the highest weight and length of the plants were recorded for the tomato plants growing at a density of nine plants per growing container. In the remaining combinations, in which three and six tomato plants were planted, the growth and yield were lower.

In the present study, strains of bacteria beneficial to plants were selected, and three microbial consortia were developed with properties to stimulate the growth and protection of romaine lettuce plants in aquaponic cultivation. The selected bacterial strains belonged to the genera *Klebsiella* spp., *Nocardia* spp., *Rhodococcus* spp., and to the species *Paenibacillus polymyxa*. So far, in spinach aquaponic crops, a positive effect on the growth and yield of spinach has been observed when the applied bacteria were from the genera *Lactobacillus*, *Bacillus*, *Nitrosomonas*, and *Nitrobacter* [77]. Similarly, studies have found a beneficial effect of *Rahnella* spp. and nitrifying bacteria Nitrosomonas spp. and *Nitrobacter* spp. on the yield of lettuce *Lactuca sativa* [21,52]. The conducted research confirms the positive effect of selected bacterial strains on the growth and yield of leafy vegetables and indicates the possibility of using innovative microbial consortia in the aquaponic cultivation of lettuce.

The application of beneficial microorganisms often increases the productivity of aquaponic crops and contributes to improving the quality of the crop obtained. In the present study, the application of microbial consortia or minerals had a positive effect on the growth of romaine lettuce plants in aquaponic cultivation. The positive effect of the application of microorganisms obtained in this study also confirms the findings of other authors. Aini et al. [78] reported that the use of AMF, PGPR, or a consortium of AMF with PGPR in hydroponic cultivation increased the yield of romaine lettuce. The length and

thickness of the leaves and also the weight of the plants were greater. This was particularly evident when using a consortium that included arbuscular mycorrhizal fungi and beneficial rhizosphere bacteria. The addition of AMF and bacteria increased the uptake of minerals from the medium; however, the efficiency of their uptake depended on the concentration of the medium (measured as the electrical conductivity of the medium used). The highest efficiency of nitrogen uptake was observed after using a consortium of microorganisms and a nutrient solution with ion concentrations of 1.8 dS m$^{-1}$ and 0.9 dS m$^{-1}$; however, at a concentration of 1.4 dS m$^{-1}$, the highest efficiency of nitrogen uptake was observed in the plants inoculated with AMF only. Additionally, after the application of AMF, the highest efficiency of phosphorus and potassium uptake was noted. The application of beneficial bacteria increased the efficiency of mineral uptake from the medium compared to the non-inoculated control. Similar results were obtained by Kasozi et al. [79], testing the effect of wastewater from the production of Mozambique tilapia treated with a consortium of Bacillus bacteria on the growth of lettuce. Plants fed with the water treated with the bacteria were characterized by higher fresh and dry weight of leaves and roots, regardless of the date of cultivation. Plants grown in the water treated with the bacteria were also characterized by significantly higher concentrations of potassium, phosphorus, and zinc. Ajijah et al. [21] reported an improvement in the growth parameters of lettuce plants inoculated with Nitrosomonas europaea and Nitrobacter winogradskyi compared to plants grown in traditional hydroponics and also in an aquaponic combination not enriched with beneficial microorganisms. The plants of the scented vetiver tested in that experiment also had better growth parameters. In addition, the plants inoculated with beneficial bacteria contained more phosphorus. On the other hand, the concentration of ammonium ions was significantly reduced in the wastewater treated with the bacteria.

However, the application of beneficial microorganisms does not always contribute to increasing the yield of plants in aquaponics compared to plants fed only with wastewater from fish farming. Effendi et al. [80] tested the usefulness of romaine lettuce cultivation to purify wastewater from Nile tilapia (*Oreochromis niloticus*) farms. The authors concluded that romaine lettuce plants could be grown on water from tilapia farms without the need for additional mineral supplementation. The authors noted that the additional application of beneficial bacteria reduced the growth of lettuce compared to plants fed with wastewater only; however, the differences obtained were not statistically significant.

In the present study, an increase in bacterial activity was observed as a result of using minerals and microbial consortia. Schmautz et al. [81] found that in aquaponic crops, microbial activity depends on abiotic factors, with oxygen saturation, total organic carbon, and total nitrogen being the most influential. The present results confirm the influence of mineral nutrients on the activity of bacteria in aquaponic crop cultivation. In the aquaponic cultivation of Lactuca sativa, temporal changes in the composition of bacterial communities have been observed depending on the aerobic or anaerobic conditions of the system [82]. In the present study, increased bacterial biodiversity was observed only 48–72 h after the application of microbial consortia and minerals, but the tests were conducted under uniform aerobic conditions.

In the present study, the best growth parameters of romaine lettuce plants were obtained after supplementation of the fish farm wastewater with mineral nutrients. The greatest activity of microorganisms was also observed in this combination. In aquaculture production systems, microbial communities play significant roles in nutrient recycling, the degradation of organic matter, and the treatment and control of the disease [83]. One may assume that the increased doses of mineral nutrients resulted in greater activity of the microorganisms involved in nutrient cycling in aquaponic lettuce cultivation.

In aquaponic cultivation of spinach, plants inoculated with microbial consortia were found to have significantly higher growth and yield than the non-inoculated control, supporting the hypothesis that the use of microbial consortia can increase the yield of spinach plants [77]. The results of the present study confirm these observations and indicate the possibility of increasing the yield of lettuce plants in aquaponic cultivation by

applying consortia of beneficial microorganisms or mineral nutrients to the growth media. However, the introduction of microbiological consortia into the cultivation of romaine lettuce requires further research, focusing especially on the use of beneficial microorganisms to protect plants against pathogens easily spreading in the aquatic environment, such as Phytophthora oomycetes.

## 5. Conclusions

The application of microbial consortia or mineral nutrients had a positive effect on the growth of romaine lettuce plants in aquaponic cultivation using fish farm wastewater from the production of hybrid sturgeon (*Acipenser gueldenstaedtii* Brandt × *Acipenser baerii* Brandt). The best growth parameters of lettuce plants were obtained after the application of mineral nutrients, but the plants were characterized by the strongest symptoms of leaf tipburn and the lowest commercial value. The occurrence of tipburn is mainly associated with disturbances in the transport of calcium as a result of the rapid growth of plants fed with minerals at relatively high humidity and inadequate light parameters. The experiment was set up in a simulated aquaponic system without recirculation, which may have contributed to the tipburn occurrences in romaine lettuce. The results of our study might have been different if the experiment had been conducted with a coupled aquaponics system. The application of microbial consortia and minerals significantly increased the numbers and activity of bacteria in the culture liquids 7, 14, and 21 days after inoculation. As a result of the conducted research, four strains of bacteria belonging to the genus *Klebsiella*, *Nocardia*, and *Rhodococcus* were selected, with the properties of oxidation and growth of $NH_4^+$ ions on a mineral medium containing $NH_4Cl$ as a source of nitrogen, and two strains of bacteria belonging to the genus *Paenibacillus*, with properties inhibiting the pathogen *Phytophthora cactorum*. Selected strains of microorganisms are part of three microbiological consortia that can be used in aquaponic cultivation, contributing to the implementation of natural technologies supporting the growth and yield of romaine lettuce plants.

**Author Contributions:** Conceptualization, L.S.-P., S.K. and B.M.; methodology, L.S.-P., P.T. and S.G.; formal analysis, P.T. and S.G.; investigation, P.T., S.G., A.L. and B.M.; data curation, P.T., A.L. and B.M.; writing—original draft preparation, L.S.-P. and S.G.; writing—review and editing, A.L., B.M. and S.K.; supervision, L.S.-P.; project administration, S.K.; funding acquisition, L.S.-P. and S.K. All authors have read and agreed to the published version of the manuscript.

**Funding:** This work was supported by funding through the European Union from the European Regional Development Fund under the Smart Growth Operational Program 2014-2020 and The National Centre for Research and Development in Poland. Project title: "Plantlab—innovative system of year-round production of romaine lettuce and freshwater fish in aquaponic technology".

**Institutional Review Board Statement:** Not applicable.

**Informed Consent Statement:** Not applicable.

**Data Availability Statement:** The data presented in this study are available on request from the corresponding author.

**Conflicts of Interest:** The authors declare no conflict of interest.

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
