# Peer review of "The Influence of Consortia of Beneficial Microorganisms on the Growth and Yield of Aquaponically Grown Romaine Lettuce"

_agronomy, doi:10.3390/agronomy13020546_

Round 1

Reviewer 1 Report

Please check the file attached.

Author Response

Dear Reviewer,

The authors are very grateful for valuable comments that contribute to increasing the value of the manuscript. The authors revised the manuscript in accordance with all comments.

Remark1: Material and methods: 2.1. : I suggest that you briefly (2-3 sentences) describe the procedure how you determined the inhibitory properties.

Response: In Materials and Methods, we described the procedure for determining the properties of bacterial strains to inhibit the growth of P. cactorum: „The production of secondary metabolites toxic to phytopathogenic fungi was assessed by the double-culture method. Plates with agar media (potato dextrose agar and soil agar) were inoculated with a culture of P. cactorum in the centre and with the test bacteria at the edge. After inoculation, the plates were incubated for 14 days at 26°C. Bacterial isolates that inhibited the growth of P. cactorum were judged as producing toxic secondary metabo-lites against the pathogen.“ (l. 158-164).

Remark 2: Table 1.: please put the numbers in superscript/subscript (e.g. PO4-3).

Response: In Table 1, numbers are entered in superscript/subscript.

Remark 3: Line 222/223: is the equation correct?

Response: The equation has been corrected (l. 233).

Remark 4: Table 3.: superscript were necessary (e.g. cm2)

Response: Superscripts in Table 3 have been supplemented.

Remark 5: Line 293: superscript ml-1

Response: Superscript has been added (l. 304)

Reviewer 2 Report

The manuscript is well-written and presented, some comments are here and in the attached pdf needs to be considered before acceptance:

- The literature review are too much (81). can be focused on the most recent.

- The statistical analysis section should be with more details and should be mentioned in all figures and tables.

- the statistical analysis letters, should be matching with the standard error differences.

- Standard error should be add to all data in tables

Author Response

Dear Reviewer,

The authors are very grateful for valuable comments that contribute to increasing the value of the manuscript.

We provide responses to the Reviewer's comments:

Remark 1: The literature review are too much (81). can be focused on the most recent.

Response: In our opinion, the references should not be changed.

Remark 2: The statistical analysis section should be with more details and should be mentioned in all figures and tables.

Response: Statistical data has been corrected in accordance with the reviewer's comments

Remark 3: - the statistical analysis letters, should be matching with the standard error differences.

Response: Statistical analysis letters follow standard error differences.

Remark 4: - Standard error should be add to all data in tables

Response: Standard error has been added to all data in tables.

Remark 5: Add this citation: (Ahmed et al., 2021) Ref: Ahmed, Z.F.R.; Alnuaimi, A.K.H.; Askri, A.; Tzortzakis, N. Evaluation of Lettuce (Lactuca sativa L.) Production under Hydroponic System: Nutrient Solution Derived from Fish Waste vs. Inorganic Nutrient Solution. Horticulturae 2021, 7, 292. https://doi.org/10.3390/horticulturae7090292

Response: The citation of this publication has been added.

Remark 6: Citation: Albadwawi et al., 2022 Ref: Albadwawi, M.A.O.K.; Ahmed, Z.F.R.; Kurup, S.S.; Alyafei, M.A.; Jaleel, A. A Comparative Evaluation of Aquaponic and Soil Systems on Yield and Antioxidant Levels in Basil, an Important Food Plant in Lamiaceae. Agronomy 2022, 12, 3007. https://doi.org/10.3390/agronomy12123007

Response: The citation of this publication has been added.

Remark 7: Table 3 - add the standard error, put a foot note for the table explaining the statistical analysis test, replicate number

Response: The table has been corrected.

Remark 8: Figure 1 - revise the statistical letters that show significance. the standard error s are not overlaps, the letters should not be similar. The letter a should be given to the highest mean value and b for the next and so on

Response: Figure 1has been corrected. The letter “a” denotes the lowest value, which is consistent throughout the article.

Remark 9: Table 4 - same here standard error should be added

Response: The table has been corrected.

Remark 10: should be c not a then the bigger value should be a

Response: In our opinion, the letters should not be changed.

Remark 11: Too much reference for an experimental article, focus on the recent citations

Response: In our opinion, the references should not be changed.

Reviewer 3 Report

1-The title of the manuscript is very long, please, make it short and clear.

2- Keywords are Ok and it does not need any changes.

3- Both Introduction and Material and Methods are OK, and they do not need any changes.

4- Table 2 and 3 should be designed according to the journal s format.

5- The discussion part is OK, but conclusion should be increased and give more information from the main part of the manuscript.

6- Authors should write DOI for all References.

Author Response

Dear Reviewer,

The authors are very grateful for valuable comments that contribute to increasing the value of the manuscript.

We provide responses to the Reviewer's comments:

Remark1: The title of the manuscript is very long, please, make it short and clear.

Response: In our opinion, the tittle of the manuscript should not be changed.

Remark 2: Table 2 and 3 should be designed according to the journal s format.

Response: Tables 2 nad 3 have been corrected according to the journal format.

Remark 3: The discussion part is OK, but conclusion should be increased and give more information from the main part of the manuscript.

Response: The conclusions were extended with more information on the occurrence of tipburn and the use of beneficial microorganisms in aquaponic technology (l.431-446).

Remark 4: Authors should write DOI for all References.

Response: DOI numbers have been added to references.

Reviewer 4 Report

This study investigated the effect of different supplements (three types of beneficial bacteria and mineral nutrients) into fish fam wastewater on the growth of Romaine lettuce. Biomass and marketable values were compared. The experiment was well designed. Here are some suggestions to improve this manuscript.

1.      L168: Please indicate the product manufacturing information.

2.      L 185: In this treatment, N-NO3 is low, while Ca is too high. Please explain the reason using this formula or indicate the reference.

3.      Suggest the author review and explain why the treatment with supplemental minerals showed the highest tipburn rate.

4.      In Conclusion session, suggest the author points out that this experiment setting is not using recirculating aquaponic system, but a simulation experiment. Results may different with coupled aquaponics system, which may explain the tipburn issue, etc.

Author Response

Dear Reviewer,

The authors are very grateful for valuable comments that contribute to increasing the value of the manuscript.

We provide responses to the Reviewer's comments:

Remark 1: L168: Please indicate the product manufacturing information.

Response: The following text has been added to the manuscript: „an outdoor free-standing container (Weldon, Brzezówka, Poland) with dimensions 6.0 × 2.6 × 3.2 m adapted to a phytotron by BIOSELL (Warsaw, Poland) fitted with two two-shelf racks“ (l. 172-174). Information about the manufacturer of rockwool has also been added (l. 176-177)

Remark 2: L 185: In this treatment, N-NO3 is low, while Ca is too high. Please explain the reason using this formula or indicate the reference.

Response: The nutrient composition for indoor romaine lettuce cultivation was selected based on our previous research.: Matysiak, B.; Kaniszewski,S.; DyÅ›ko, J.; Kowalczyk, W.; Kowalski, A.; Grzegorzewska, M. TheImpact of LED Light Spectrum on the Growth, Morphological Traits, and Nutritional Status of ‘Elizium’ Romaine Lettuce Grown in an Indoor Controlled Environment. Agriculture2021, 11, 1133. https://doi.org/10.3390/agriculture11111133. This publication is cited at this point in the manuscript (l. 196).

Remark 3: Suggest the author review and explain why the treatment with supplemental minerals showed the highest tipburn rate.

Response: The occurrence of tipburn is mainly associated with disturbances in the transport of calcium as a result of the rapid growth of plants fertilized with minerals at relatively high humidity and inadequate light parameters. (Matysiak, B.; Kaniszewski, S.; DyÅ›ko, J.; Kowalczyk, W.; Kowalski, A.; Grzegorzewska, M. The impact of LED light spec-trum on the growth, morphological traits, and nutritional status of ‘Elizium’ romaine lettuce grown in an indoor controlled environment. Agriculture 2021, 11, 1133. https://doi.org/10.3390/agriculture11111133)

Remark 4: In Conclusion session, suggest the author points out that this experiment setting is not using recirculating aquaponic system, but a simulation experiment. Results may different with coupled aquaponics system, which may explain the tipburn issue, etc.

Response: The following text has been added to the manuscript: „The occurrence of tipburn is mainly associated with disturbances in the transport of calcium as a result of the rapid growth of plants fertilized with minerals at relatively high humidity and inadequate light parameters. The experiment was set up in a simulated aquaponic system without recirculation which may have contributed to the tipburn occurrences in romaine lettuce. The results of our study might have been different if the experiment had been conducted with coupled aquaponics system“ (l. 431-437).
